# More than ticking boxes: Training Lyme disease education ambassadors to meet outreach and surveillance challenges in Québec, Canada

Karl Forest-Bérard[1]*, Marion Ripoche[1], Alejandra Irace-Cima[1,2], Karine Thivierge[3,4,5], Ariane Adam-Poupart[1,2,5]

1 Direction des risques biologiques et de la santé au travail, Institut national de santé publique du Québec (INSPQ), Montréal, Québec, Canada, 2 École de Santé Publique de l'Université de Montréal (ESPUM), Université de Montréal, Montréal, Québec, Canada, 3 Laboratoire de santé publique du Québec (LSPQ), Institut national de santé publique du Québec (INSPQ), Montréal, Québec, Canada, 4 Institute of Parasitology, Faculty of Agricultural and Environmental Sciences, McGill University, Montréal, Québec, Canada, 5 Groupe de recherche en épidémiologie des zoonoses et santé publique (GREZOSP), Faculté de médecine vétérinaire, Université de Montréal, Saint-Hyacinthe, Québec, Canada

* karl.forest-berard@inspq.qc.ca

## Abstract

Lyme disease (LD) is an emerging public health threat in Canada, associated with the northward range expansion of the black-legged tick (*Ixodes scapularis*). To address this, public health authorities have been carrying out surveillance activities and awareness campaigns targeting vulnerable populations such as outdoor workers. Implementing these measures is time-consuming and resource-intensive, prompting the assessment of alternatives. Our goal was to evaluate the feasibility and implementation of a training-of-trainers-inspired approach in raising awareness about LD risk and prevention among workers and general population, as well as to evaluate its potential to contribute to provincial LD surveillance efforts. We trained a group of workers from publicly-accessible outdoor parks of the province of Québec to become "LD education ambassadors". Ambassadors were trained to raise tick and LD awareness, share information on preventive measures in their respective communities, and lead tick sampling activities using a standardised protocol similar to that used by Public Health authorities. Ambassador-led outreach activities, public reach, sampling activities and collected ticks were documented, as well as ambassadors' satisfaction with the training using forms and semi-structured interviews. In total, 18 ambassadors from 12 organizations were trained. Between June and September 2019, they led 28 independent outreach activities, reaching over 1 860 individuals (from occupational and general public settings) in seven public health units. Ambassadors led 28 tick samplings, together collecting 11 *I. scapularis* ticks. This study suggests that an adapted training-of-trainers is a feasible approach to raising tick and LD risk awareness among Québec outdoor workers and public. Trained ambassadors have the potential of reaching a large portion of the population visiting or working in outdoor parks while also providing much-needed outreach regarding

**Data Availability Statement:** All relevant data are within the paper and its Supporting Information files.

**Funding:** Our work was supported by a grant to AAP from the Infectious Disease and Climate Change Fund (IDCCF) from the Public Health Agency of Canada (PHAC). The funders had no role in study design, data collection and analysis, decision to publish, or preparation of the present manuscript.

**Competing interests:** The authors have declared that no competing interests exist.

**Abbreviations:** CS, Citizen Science; INSPQ, Institut national de santé publique du Québec; LD, Lyme disease; NCC, Nature Conservancy of Canada; PH (U), Public health (unit); ToT, Training-of-Trainers.

risk and prevention. Pushing this concept further to include other types of workers and jurisdictions may contribute to national LD surveillance efforts.

## Introduction

With the exception of a few native species, Canada's environment has historically been unsuitable for hard ticks, as typical winter conditions are detrimental to their development, survival and spread. However, shifting climate patterns along with associated ecological changes and eroding land-use may be easing the way for non-native ticks to colonize and settle into new regions north of the Canadian-American border. Most likely introduced and dispersed by migratory animals they feed on (mostly avian species, such as migratory songbirds), these adventitious ticks are progressively expanding their geographic distribution range into upper latitudes [1, 2]. Such is the case of the black-legged tick, *Ixodes scapularis* (Acari: Ixodidae, Say 1821; the term 'tick' only refers to *I. scapularis* hereafter), the main northeastern American vector of Lyme disease (LD, or Lyme Borreliosis) [3, 4].

LD is now the most commonly reported vector-borne disease in Canada [5–7]. Québec is among the Canadian provinces with the most LD cases declared annually. The first locally-acquired case in Québec was confirmed in 2006 and numbers have been growing ever since, rocketing from 2 cases in 2008 to 500 cases in 2019 (confirmed or suspected cases) [8–11]. As environmental suitability increases along with climate warming, continued northward range expansion of *I. scapularis* is expected in the upcoming decade [3].

To address this important emerging issue, public health (PH) authorities in the province are leading campaigns to raise awareness of the public and conducting surveillance activities to keep track of evolving risk-areas. Awareness-raising campaigns mostly consist of disseminating the information on preventive measures via websites, media advertisements, pamphlets and posters distributed in public parks, community centres, hospitals, etc. [12–15].

Due to limited available resources, these campaigns usually target the general population and rarely reach outdoor workers, even if some of them are typically more exposed to tick bites (forestry or agricultural workers for instance) [16–19]. Surveillance activities involve the gathering of information on human cases of LD and collected ticks. The later are collected passively (i.e. reported tick bites on humans and animals) or actively (i.e. in the field using a standardised flannel dragging protocol). Combined, these human, animal and tick data allow the creation of a provincial LD risk map, which guide decision-makers, and help clinicians for the diagnosis and treatment of patients [9]. Active surveillance produces the best dataset and is considered the gold-standard method to identify LD risk areas [20]. However, it is very resource-intensive, requiring the involvement of several partners and a trained workforce available during tick season. Financial constraints and the sheer size of Québec's territory (1,667,712 km$^2$) also make active surveillance challenging to implement. Alternative methods to further improve the geographic coverage and efficiency of this activity are always sought [21].

This project aimed at addressing some of the challenges associated with LD outreach and surveillance in Québec by exploring an adapted version of the "training of trainers" teaching methodology (ToT). This approach relies on the idea that every student can become a teacher if given the proper tools and time. A single PH professional may convert less-experienced individuals into new trainers, who can subsequently pass knowledge on to others in their vicinity [22–24]. To our knowledge, multiple ToT success stories have recently been documented in

the health intervention literature, although never specifically for tick-borne diseases in Canada [25–27].

Our goal was to evaluate the feasibility and implementation of an adapted ToT method in raising awareness about LD risk and prevention among Québec workers and the general population, as well as its potential to contribute to provincial LD surveillance efforts. More specifically, we selected and trained workers or administrators from various publicly accessible natural parks and conservation organisations to become Lyme education instructors, or "prevention ambassadors", in charge of raising tick awareness, sharing information on LD and preventive measures, as well as leading ticks sampling activities in their respective communities.

## Materials and methods

This project was carried out from January to December 2019 in the province of Québec, and was divided into five stages: Conceptual development of the training; Ambassador selection and enrolment; Ambassador training; Ambassador outreach and tick sampling activities; Analysis and evaluation.

### Training concept development

Training content was developed based on the idea that LD prevention ambassadors had to be able to communicate accurate information to the public and to workers, as well as to conduct tick sampling activities using a standardized protocol, with remote assistance if needed. To achieve this, 3h in-person workshops were designed as a single session covering both theory and practice.

### Theoretical component

Content was elaborated using PH recommendations issued by federal and provincial instances [12–15, 28–33], supplemented by online material and existing literature [24, 34]. It was adapted for a public with various levels of literacy, favouring a graphics-rich and jargon-free design. The selected content was reviewed and validated by a cross-disciplinary committee composed of consulting human and veterinarian physicians, academics, biologists, parasitologist, clinical nurse and communication counsellor.

The first half of the training consisted of a 90-minute lecture covering five topics: project context, pedagogical tools and outreach examples; basic concepts of tick biology; LD clinical presentation; preventive measures and tick removal procedure; introduction to sampling methodology for tick surveillance. The lecture was designed to be interactive, encouraging group discussions and including tick specimen identification and multimedia quiz (gamification). Each participating ambassador also received a turn-key training kit, which included a bilingual workbook (available in French or English to limit possible language barriers) meant as a stand-alone document that ambassadors could rely on and use after the workshop. The contents of the training kit provided to each ambassador are shown in **Fig 1A**.

### Practical component: Tick sampling protocol

The second half of the training consisted of a 90-minute fieldwork activity, during which ambassadors had the opportunity to experience first-hand the dragging protocol under the instructor' supervision. This activity was performed in the vicinity of the workshop in tick habitat, and mostly served as a practice run (**Fig 1B**). The tick sampling procedure was adapted from the standardized protocol currently in use in Québec's provincial integrated surveillance program [9] (**Fig 2**), slightly simplified in order to reduce unnecessary exposure to tick bites

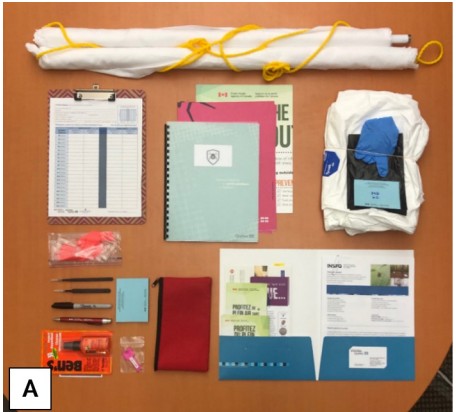 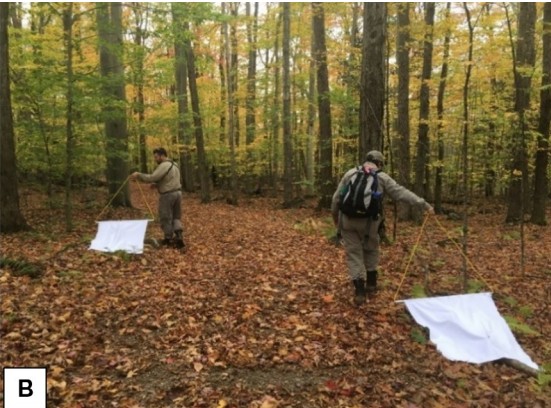

**Fig 1. Reference pictures of the project.** Photographs depict (**A**) the content of the ambassador training kit provided at workshop (bilingual workbook, outreach material, USB key with project forms and additional material, informed consent, and the necessary tools for tick sampling, including white flannels, fine-tipped tweezers, insect repellent, 2mL ethanol-vials, Tyvek® suits and personal protection equipment), and (**B**) the flannel-dragging session conducted by newly-formed ambassadors during a workshop's fieldwork segment.

and facilitate implementation. The ambassador starting kit included personal protective equipment to be worn during sampling activities. Briefly, each sampling consisted of dragging a 1 m$^2$ white flannel cloth attached to a horizontal pole across the forest floor alongside the trail of the selected area (off-path, within a 5 m margin). Sampling was performed on a total distance of 2 000 m, subdivided into four 500 m transects (two on each side of the trail). Samplers stopped every 25 m to inspect the cloth for ticks before resuming. Samplings were done on a voluntary basis only. Protocol required samplings to be performed in teams of at least two persons for security and efficiency purposes [9].

## Ambassador selection, enrolment and consent to participate

To find candidates, we partnered with Nature Conservancy of Canada (NCC), a private non-profit organization whose main focus is conservation of natural areas and wildlife through long-term securement, management and restoration of a variety of properties at national level [35]. Potential ambassadors were identified among NCC's employees, administrators and collaborators, and selected according to the geographic locations of their workplace (endemic regions prioritized), previous experience in outreach and fieldwork, general interest and availability to participate (attending a three-hour training workshop, plus a flexible amount of time during the summer between June-September to organise and lead outreach and tick sampling activities).

Prior to enrolment, all participants provided written informed consent to take part in this project. They were informed of its aims, methods and sources of funding, possible conflicts of interest and institutional affiliations of the researchers, as well as anticipated benefits and potential risks of their participation to this study. They were informed of their right to refuse to participate or to withdraw consent to participate at any time without question or reprisal.

## Ambassador training

Three workshops led by a professional scientific advisor from INSPQ were given to ambassadors in May 2019, at the beginning of tick season and before tourism high season, when workers are the busiest. Workshops were given in French or English in three different PHUs (Montérégie, Outaouais, Chaudière-Appalaches) located within or in proximity of emerging

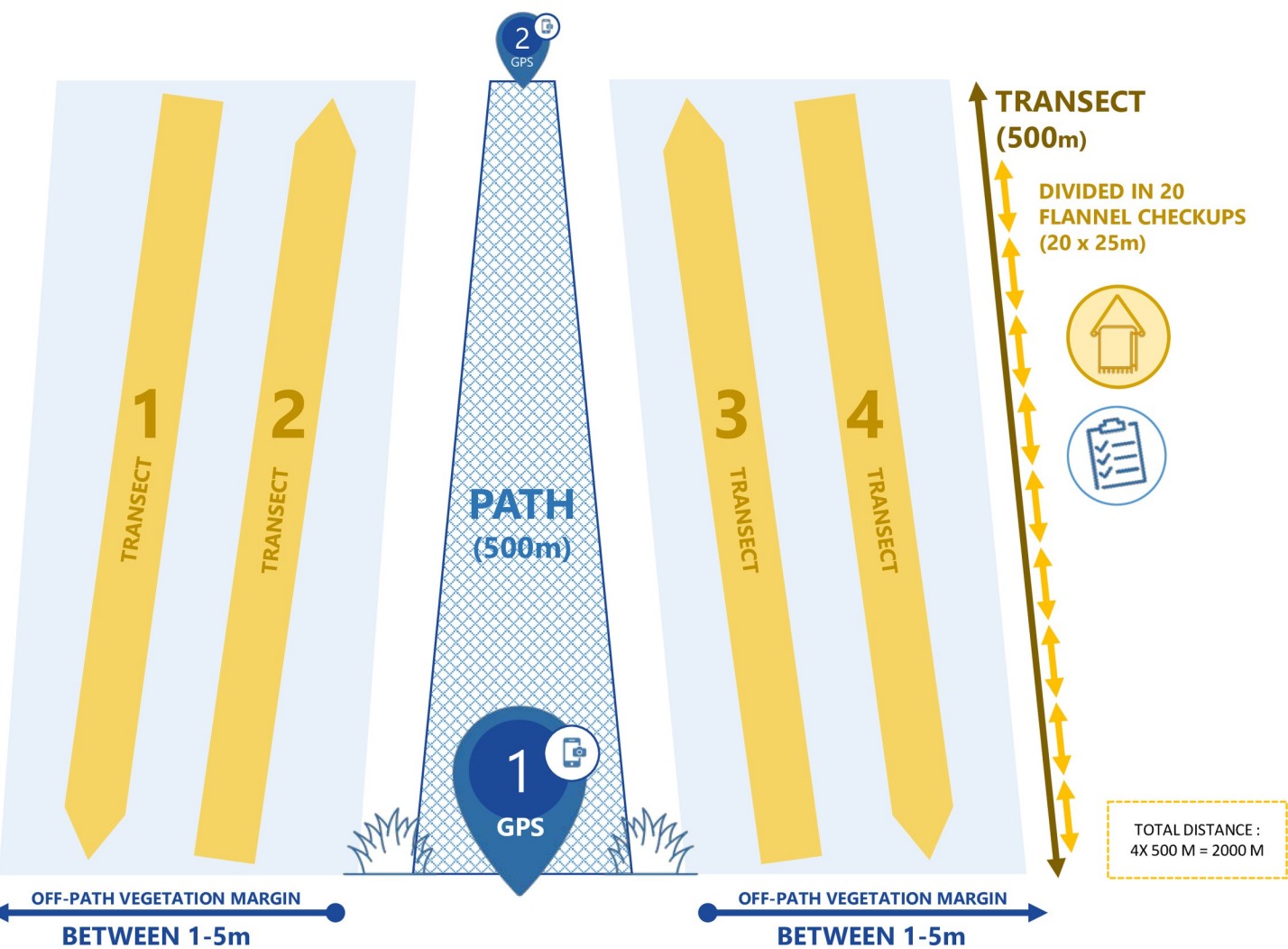

**Fig 2. Schematic representation of the tick sampling protocol given to ambassadors.** Four 500 m transects (yellow) are sampled by dragging a 1 m$^2$ white flannel cloth on the ground, within a 1–5 m margin (light blue area) alongside the chosen trail path (total distance 2000 m per sampling session, representing approximately 3 person-hours). Flannel is inspected for ticks every 25 m, and GPS location recorded at each end. Ticks found are transferred in ethanol-filled tubes and kept until identification at the lab by specialists (adapted from Adam-Poupart, et al. 2016) [9].

LD areas [28] in NCC's partners buildings. The theoretical part of the workshops was organised indoors, while tick sampling was conducted outside, where ecological characteristics for tick habitats were met (i.e., tall grass, areas with leaf litter, woodland trails, primarily deciduous or mixed-deciduous forests, etc.).

### Ambassador outreach and tick sampling activities

Following the workshop, ambassadors were invited to organise outreach and sampling activities in their respective organisations from June to September 2019. Ambassadors were asked to fill out an electronic form after each activity broadly describing the activity in terms of time and place, turnout, type of activity and number of ticks collected. Sampling sites were selected by ambassadors themselves, based on site suitability and availability, ambassador's experience and interest, and with guidance from PH professionals from our research team when needed. No firm minimum was required, but ambassadors were invited to lead as many activities as

possible, suggesting they completed at least 2 samplings (2x 2000m). Three follow-up calls were scheduled with ambassadors during the activity period to keep participants mobilized and motivated, make sure they had everything they needed to go forward with activities, answer questions, and keep track of completed/upcoming activities.

### Analysis and evaluation

Following workshops, newly-trained ambassadors were asked to evaluate the training in a short, anonymous survey online (using the SurveyMonkey® platform). A copy of the survey questionnaire and a portrait of overall answers (weighted averages per question) can be found in Supporting information. Implementation of the ToT-inspired approach was broadly evaluated by measuring the number of outreach activities organised per ambassador and overall, number of trainees participating per outreach activity and overall, number of PHUs (territory area) covered by outreach and tick-sampling activities, and number of *I. scapularis* ticks collected per sampling activity and among those, number of *B. burgdorferi*-infected *I. scapularis* ticks.

In addition, ambassador feedback on the ToT-inspired approach was gathered through semi-structured group interviews of 30 minutes, qualitatively covering general appreciation, usefulness of the training, perceived autonomy, positive elements of the project as well as areas its shortcomings. These comments will be used to guide improvements for future initiatives. Finally, an anonymous inbox was also set up using the SurveyMonkey® platform allowing ambassadors to provide personal or sensitive comments that could be used to improve the training.

### Tick data: Species identification and pathogen detection

Ambassadors were provided with 2 mL ethanol vials to collect specimens and addressed/stamped envelopes to mail them to the *Laboratoire de santé publique du Québec* (LSPQ) for identification. We used standard taxonomic keys [36] to identify the species, sex and life stage (larval, nymph, adult) of the ticks collected by drag sampling. Confirmed nymphal and adult *I. scapularis* ticks were screened at the national microbiology laboratory (NML) for evidence of *B. burgdorferi* using real-time PCR as previously described (total genomic DNA screened targeting the 23S ribosomal ribonucleic acid gene of *Borrelia spp*) [37, 38]. All *Borrelia spp*.-positive samples were subsequently tested for *B. burgdorferi* using a confirmatory outer surface protein A (ospA) assay. As *I. scapularis* is a competent vector of other pathogens of PH concern in Canada, samples were also tested for *Borrelia miyamotoi* [37], *Anaplasma phagocytophilum* [39], *Babesia microti* [40], *Borrelia mayonii* and deer tick virus (Powassan virus, lineage II) [41, 42], according to methods already documented in the literature.

### Ethics statement

This work was conducted in accordance with the principles expressed in the Declaration of Helsinki. According to article 2.5 of the Government of Canada's Tri-Council Policy Statement on Ethical Conduct for Research Involving Humans, program evaluation activities do not constitute research and do not fall within the scope of research ethical board review. Considering that no confidential information was collected nor analyzed, that training and personal protective equipment was provided, and that informed consent was required from each participant prior to any activity, the project was exempted from ethical review by the INSPQ's general secretariat and the project granting agency (Public Health Agency of Canada). Nevertheless, our informed consent form was reviewed for completeness and adequacy by a member of INSPQ' ethics board (*Comité d'éthique de santé publique*, CESP). All efforts

have been made to protect participants' privacy, security, and confidentiality. Also note that, while focusing on LD, other tick-borne diseases also present in the province were briefly mentioned during training. The preventive measures against *Ixodes scapularis* bites is also valid for other tick species, such *D. variabilis*, *A. americanum*, *H. leporispalustris*, and *I. cookie*. The risk of other natural risks involved when doing fieldwork (e.g. wild animals, mosquitoes, skin-irritating plants, heat shock, etc.) were not formally addressed during workshops but professional backgrounds and prior field experience was also taking into consideration when selecting ambassadors. No animal or environmental certification was needed for this type of activity. Finally, all our work was designed with inclusivity and diversity in mind at each step of the process.

## Results

### Ambassador characteristics

Eighteen ambassadors were enrolled to be trained in workshops held in May 2019 (11 women and 7 men, between 25 and 60 years of age). Trainees belonged to 12 different associations: four came directly from NCC, and the remaining 14 from its partners in public or private organisations working towards wildlife or environment conservancy, such as watershed management committee, nature interpretation center, regional environment council, volunteer groups, etc. Ambassadors were mainly biologists, wildlife technicians, project coordinators, executives, students or volunteers. Experienced outdoor workers (well-equipped to deal with potential risks out in the field) were recruited first. Most participants already had some basic knowledge about LD and ticks, and some of them had prior outreach experience (ex. park naturalists, interpretive employees). In total, ten ambassadors came from Montérégie/Estrie, five from Outaouais, two from Chaudière-Appalaches and one from Mauricie-Centre-du-Québec.

### Ambassador-led outreach activities

Sixteen of the eighteen ambassadors trained in May were active during the activity period, from June to September. A total of 28 different outreach activities were organised by fifteen of these ambassadors. On average, two activities were conducted per ambassador, ranging between zero and eight activities per person.

Using the tools and knowledge provided during workshops, ambassadors were given free rein to design and adapt outreach activities depending on their availability, location and organization. As a result, they shared the information on LD and preventive measures with colleagues, partners and clientele from their organizations in a variety of ways. These activities can be divided in two main categories: distribution of awareness-raising material (such as posters and pamphlets in workplaces, convenience stores and municipal offices, as well as informative emails and newsletters), and the organizing of free events targeting the general public or groups at-risk. Examples of such events include evening conferences, citizen discussion groups and tick identification workshops. They also include information sessions added to already-scheduled activities, for example, in the context of outdoor sports, educational or conservation activities, hunting clubs, bird/wildlife watching clubs, mountain bike clubs, summer camps, school trips, and in the context of volunteer/staff training. A few interviews with local newspapers and radio stations were also given.

While turn-out varied, these activities directly reached overall a minimum of 1 860 citizens, of whom about 10% were contacted at their workplace. Outreach activities were led in seven different PHUs, extending beyond the ones where workshops were initially held, namely in Capitale-Nationale, Chaudière-Appalaches, Montréal, Mauricie-Centre-du-Québec, Outaouais, Estrie, and Montérégie, the last four being endemic areas with significant LD risk as

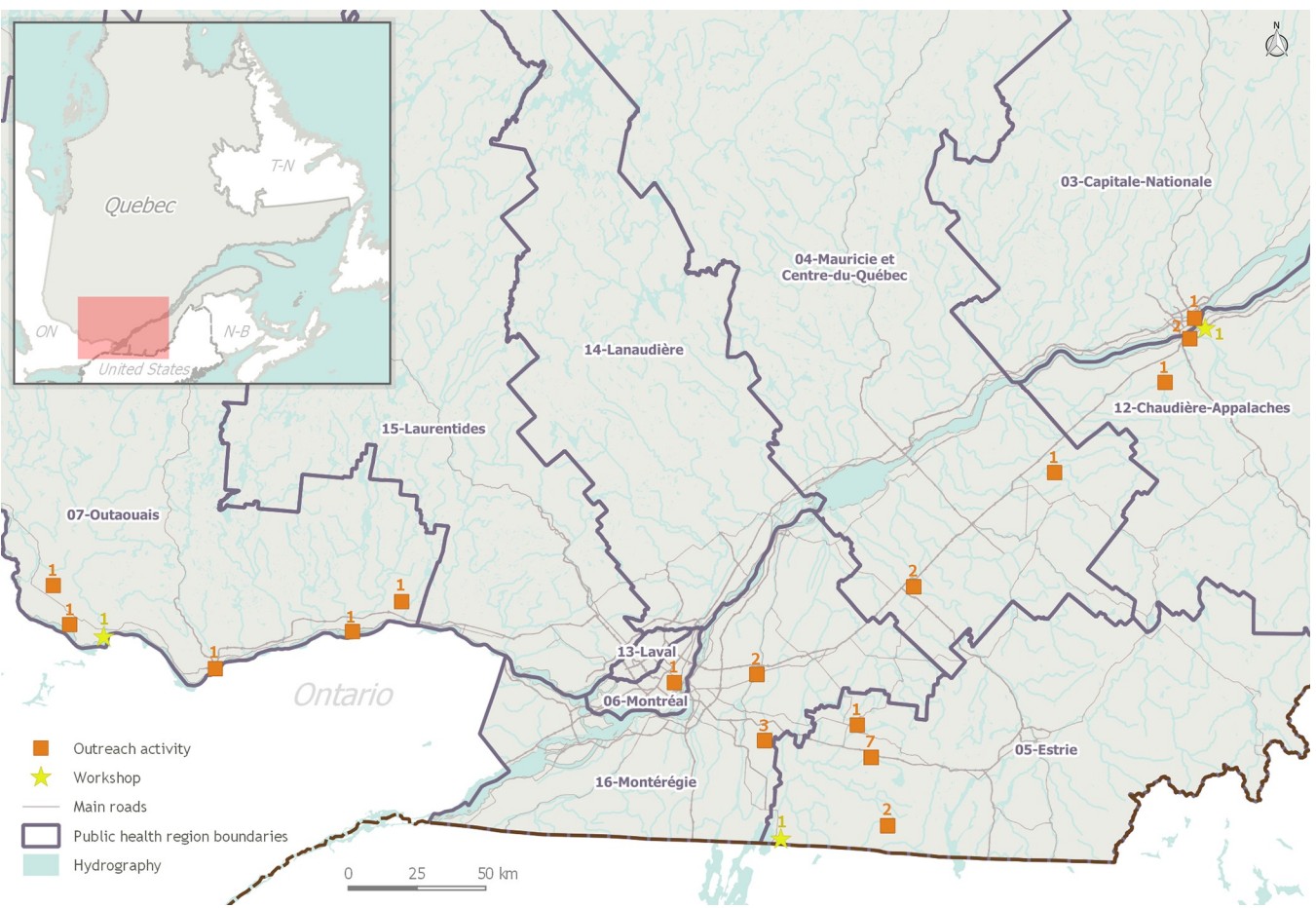

**Fig 3. Spatial distribution and amount of training workshops and ambassador-lead outreach activities across the province.** Workshops (yellow, n = 3) were held in May 2019 at three locations. From June to September 2019, newly-trained ambassadors lead outreach activities (orange, n = 28) in seven PHUs from the southern portion of the province, spanning roughly 450 km from East to West, and 210 km North-South. Activities were organised at 16 different locations (either brand new or annexed to already-existing events). Note: This map was created using open-access geographic layers produced by the *Gouvernement du Québec.*

assessed by PH authorities [28]. Some activities were ongoing, while others were repeated occasionally during the season. The seven PHUs covered the southern portion of the province, spanning roughly 450 km from East to West, and 210 km North-South. The territory covered represents only but a fraction of the province's total area but encompasses its most populous urban centers and main endemic areas. Ambassador training workshops and outreach activities are mapped in **Fig 3**.

## Ambassador-led tick sampling activities

Between June and September 2019, 11 ambassadors out of 16 led sampling activities. A total of 28 samplings were led in seven PHUs (same as outreach), but ticks were found in only three (Mauricie-Centre-du-Québec, Chaudière-Appalaches, Outaouais). Samplings were led at 16 different locations. Four samplings were positive, together yielding 11 specimens, all later identified as *I. scapularis* ticks (3 adults: 2 males, 1 female; 8 nymphs; 0 larvae). Of these, two nymphs tested positive for *B. burgdorferi* (23S, opsA) (one from Mauricie-Centre-du-Québec, one from Outaouais). No other pathogen was detected. The remaining samplings (n = 24)

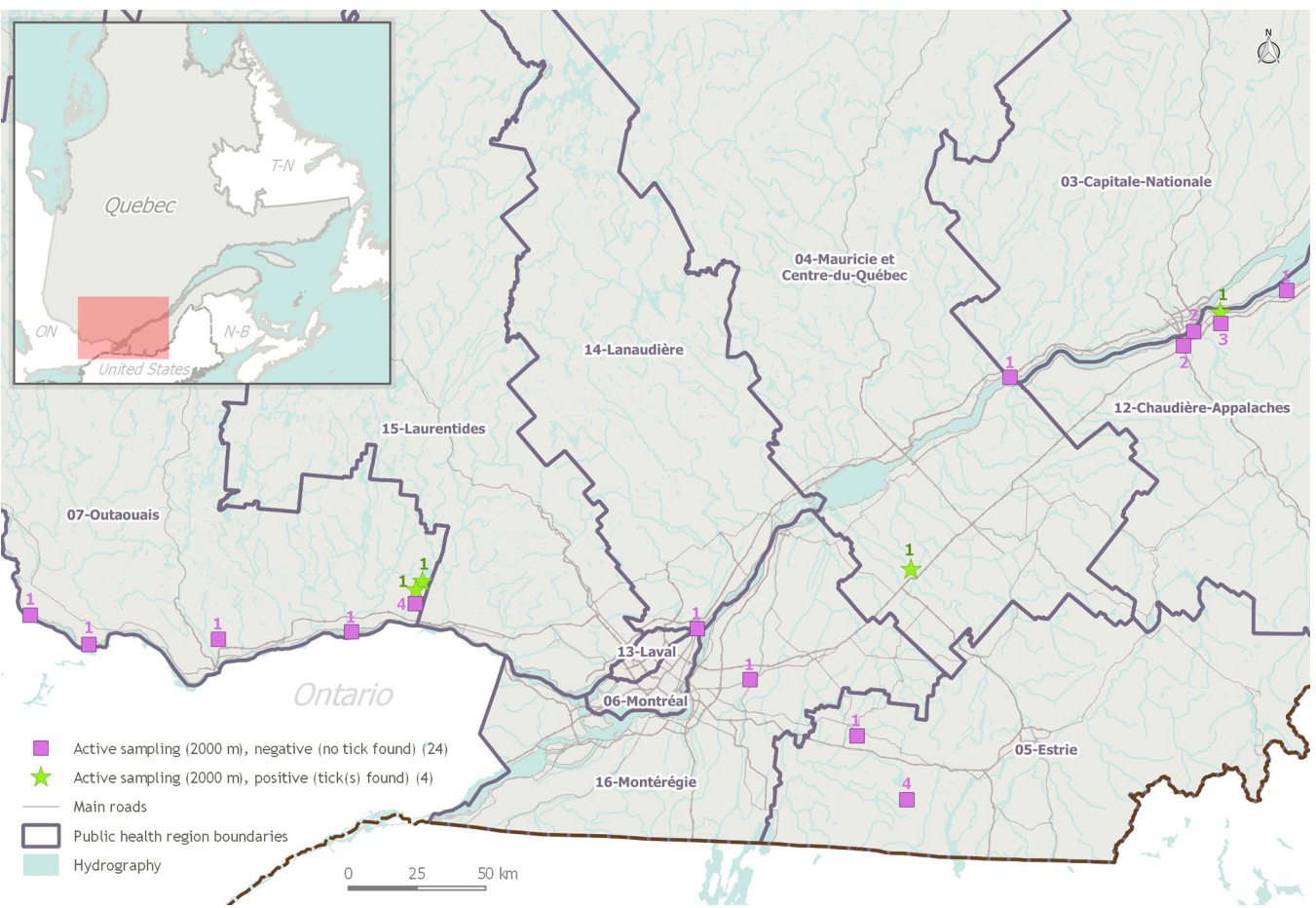

**Fig 4. Spatial distribution and quantities of ambassador-led tick sampling activities across the province.** A total of 28 tick samplings were organised by ambassadors between May and September 2019 (18 different locations, in seven PHUs). Of these, four samplings allowed tick detection (green, n = 4). The remaining samplings (purple, n = 24) were completed (2000 m sampled) but led to no detection. Note: This map was created using open-access geographic layers produced by the *Gouvernement du Québec*.

were negative (i.e. protocol completed without finding ticks). A total of 56 km of wooden area was sampled cumulatively by ambassadors. Results were shared with PH authorities in charge of surveillance as well as with participating park managers. Sampling activities during the course of this project are mapped in **Fig 4**.

## Ambassador feedback

Ambassador feedback during evaluation interviews was altogether very positive. Among the main strongpoints, ambassadors mentioned the exhaustive and high-quality content of the training, its successful "turn-key" approach simplifying every step, the project' smooth organisation and the high level of experience of the instructors. Trained ambassadors reported feeling better-informed and above all better-equipped to meet the growing LD-related challenges in their workplace. Most trainees stated they would recommend this training to others.

## Discussion

The ongoing emergence of LD throughout Canada warrants more efforts to monitor its northward progression. As part of an attempt to explore alternatives to the current LD awareness

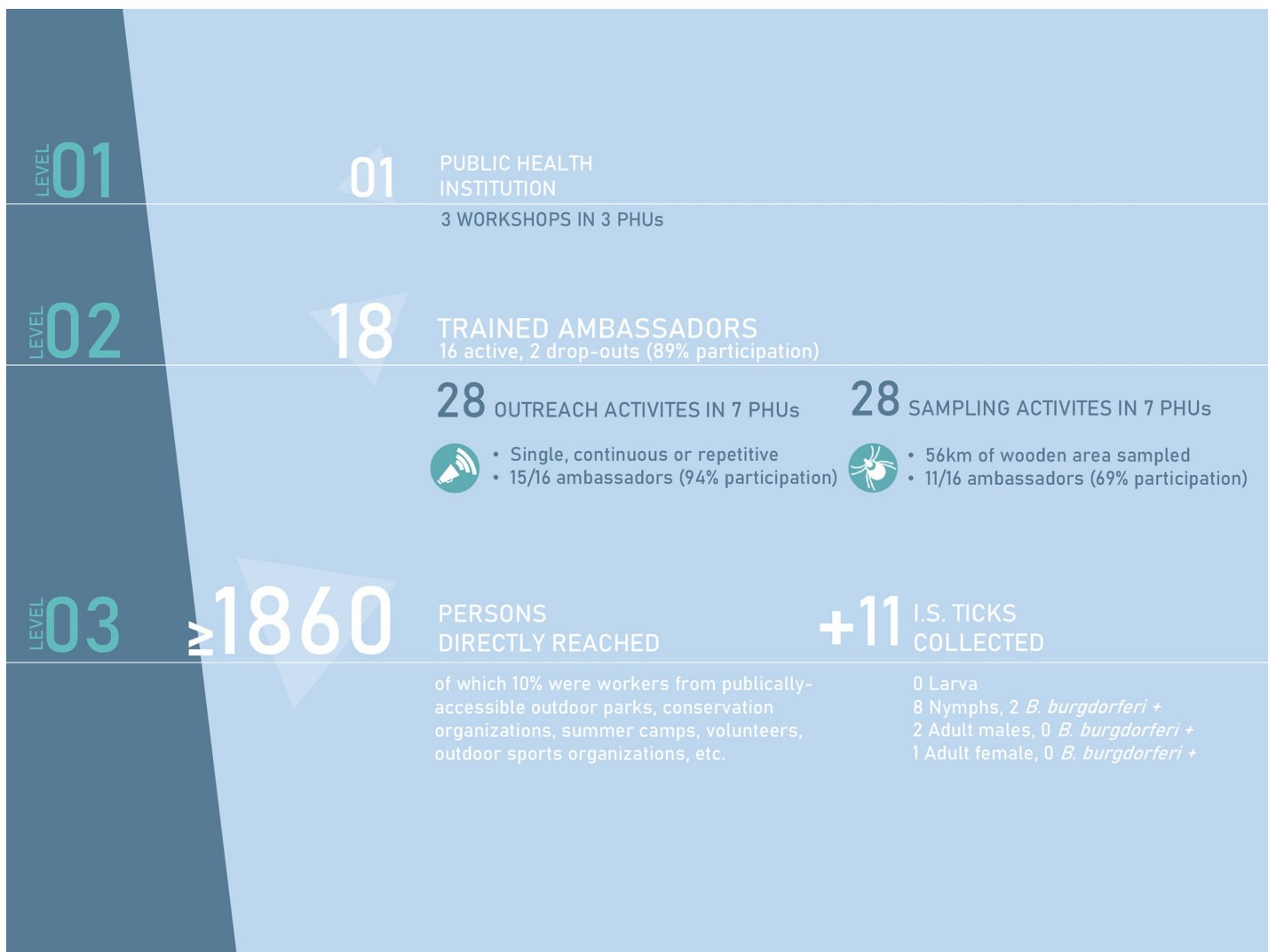

**Fig 5. Visual overview of results obtained using a training-of-trainer inspired approach to train outreach and sampling ambassadors.** Results grow increasingly from one level to the next (top to bottom), from a single PH institution leading three workshops, to 18 trained ambassadors leading 28 outreach activities and 28 tick samplings, reaching over 1 860 persons directly in seven PHUs and collecting 11 *I. scapularis* ticks.

and surveillance methods, we evaluated the feasibility and implementation of an adapted ToT-inspired approach. We found it to be a feasible and complementary approach to provincial PH efforts in raising awareness about LD risk and prevention. It also has the potential to contribute to provincial LD surveillance efforts. **Fig 5** illustrates how results of activities organized in a ToT-inspired project are amplified, growing increasingly from one level (PH institution) to the next (ambassadors and trainees).

## Facilitators and barriers to implementation

The ToT-inspired approach proved to have multiplicative effects on outreach efforts, allowing knowledge to be passed from a single PH professional to nearly 2 000 individuals throughout the south of the province, considered LD high-risk areas. This number is probably underestimated as it excludes the individuals reached through open activities such as information stands, newsletters, public interviews, etc. In addition, these results do not take into

consideration repercussions the network might have in the future, as ambassadors can keep disseminating the preventive information, either by sharing it with their colleagues and clientele in the coming years, or by training new trainers themselves.

Partnering with a private non-profit organisation such as NCC proved to be a winning combination. The quality and reach of its network coupled with the interest and field experience of the ambassadors were facilitating factors in implementing the ToT-inspired approach. The project also highlighted outdoor workers' eagerness and ability to participate in outreach and tick sampling activities. This concurs with results obtained from a province-wide 2017 online survey of outdoor park administrators, which highlighted workers' strong interest in receiving information about LD preventive measures and tick sampling (INSPQ 2017, unpublished data). As shown by ambassadors' enthusiasm, the project clearly addressed a need in terms of prevention and training, not only for vulnerable outdoor workers, but also for the general population frequenting their organisations. Such interest translates into a certain responsiveness to local concerns, which in turn could lead to higher data resolution at a regional scale. In addition, accessing NCC's private properties allowed us to survey areas never before sampled by PH authorities, which has interesting advantages to complement the surveillance datasets.

Two of the eighteen ambassadors originally recruited and trained failed to carry out outreach and sampling activities, citing professional obligations as the main barrier. There was also considerable variability in the amount of activities ambassadors carried out individually, ranging from none to eight activities per ambassador. Lack of human resources, financial constraints and lack of time were cited as their main barriers. This is understandable since ambassadors worked voluntarily, adding outreach and sampling activities to their regular professional duties. Including financial incentives could help overcome the barrier to ambassador participation while still keeping the ToT-inspired approach a relatively inexpensive option, at least compared to the classic LD surveillance model currently in place. Involving ambassadors more closely and getting them to train new staff themselves could also help preserve mobilization. Shortening the sampling protocol (covering shorter distances but including multiple visits per site or rotating samplers), organising trainings earlier in the year when workers are more available, sharing documents beforehand to cover some ground and limiting the time spent answering questions during workshops could also contribute to alleviate time investment and encourage participation.

## Strengths and limitations of the ToT approach

Workers from publicly-accessible outdoor spaces are at particularly high risk of exposure to *B. burgdorferi*-infected *I. scapularis* ticks given their recurring exposure to tick habitat [43, 44]. A reform of the occupational health and safety regime in Quebec is currently underway [45] but at the moment many of these workers do not benefit from some of the prevention mechanisms provided under the Act respecting occupational health and safety (*Loi sur la santé et la sécurité au travail*), since many of them work in a sector where only half of the prescribed prevention mechanisms are applied [46]. The ToT-inspired approach allowed us to reach workers that would not have been otherwise targeted in their workplaces in the immediate future by PH authorities based on the current provincial occupational health plan. While we do not have hard data at the moment to prove knowledge-retention, we believe these individuals benefitted from some level of added protection with the prevention information they received through ambassadors. Furthermore, training publicly-accessible outdoor park staff, who constitute a key interface with the public, has the potential to improve access of the general public to information about LD risks and prevention in a real-world setting (i.e., visiting a natural park) where this information is highly relevant.

This project also succeeded in demonstrating that tick samplings can be carried out independently by newly-trained ambassadors, who together collected 11 *I. scapularis* specimens in 28 outings. This type of sampling could complement tick data collected by the integrated LD surveillance program in Québec in two ways. Mainly organized during summer months when adult ticks are infrequently encountered, provincial surveillance samplings mostly result in the collection of larvae and nymphs. However, PH authorities are interested in identifying the presence of all three stages of *I. scapularis* at a given location over the course of one year as indicators of an established tick population, which is of relevance when defining LD endemic areas [28, 47]. In this regard, the timing of the current provincial program is not optimal, and ambassador-led samplings provide an opportunity to enhance surveillance datasets. Furthermore, while their analysis and integration might be more complex, ambassador tick data also prove to be useful in providing intensive tick monitoring of a small area where human activity overlaps with tick biology. Indeed, sites sampled by PH authorities as part of the integrated surveillance program are typically visited only once or twice per year, in areas not necessarily frequented by people. Therefore, repetitive monitoring (sites visited multiple times per year) made available by local ambassadors already on-site allows not only to characterize multiple life stages, but also generates data that is more representative (actual exposure portrait). Active surveillance samplings are known to be inefficient and dependent on numerous variables–both biotic and abiotic [48, 49]. Collecting ticks from a given location on multiple dates could contribute to moderate the potential effects these variables might have on tick detection, while also validating tick data longitudinally.

The fact that ambassadors were indeed able to collect ticks in their environment also demonstrates two things related to prevention and education: (A) that the workers recruited for our project are indeed at risk of tick bites in their occupational environment and that they do benefitted from having received educational material of bite prevention; and (B) that the sampling protocol taught is feasible, understandable and effective, but also that they understood when, where and how to find ticks, which speaks volumes in terms of education and awareness.

On a larger scale, the ToT-inspired approach could allow the establishment of a flexible network of community samplers that could provide regular collections from early spring through late fall, thus covering the entire *I. scapularis* life cycle. This would allow a thorough characterisation of tick population in endemic areas, helpful both locally (public outdoor parks) and provincially (PH authorities) [9].

Ambassador-led samplings provide a variety of other advantages compared to standard surveillance, the main ones being their inexpensiveness and their potential to cover large areas. They also allow access to never-before sampled sites. On the other hand, the overall small number of samplings in this study limits results interpretation. Low tick numbers are likely explained by ambassadors' lack of time rather than by a lack of understanding of the sampling protocol since most participants reported feeling confident in leading samplings in the project final evaluation (86,4% weighted average, shown in supplement information). It may also indicate a need for stronger post-training follow-ups.

The overall yield of ticks collected by ambassadors can appear low and insignificant (n = 11 in 28 samplings). However, taken in context with LD's actual epidemiology in Québec, and based on the regional risk-assessment criterions currently in place in the province, every collected tick has the potential to change the LD risk level of the given municipality where it was detected. In the case of this specific project, most ticks collected by ambassadors came from locations were PH authorities had already recorded a *I. scapularis* presence signal in the past. However, one of the collected ticks came from an area where such signal had not yet been documented, thus changing the risk area for that municipality from 2019 to 2020 (along with

other data from the provincial active surveillance program). This change appeared on the yearly provincial risk-assessment map and could translate into different PH interventions and recommendations for this area.

In past studies, having an unstructured methodology for tick recovery proved to be a disadvantage for data interpretation [50]. The ambassador-approach using a standardized protocol close to that used by PH authorities overcame that issue. The fact that so many tick samplings were negative (completed protocol without detecting ticks) raises some questions about adherence to the procedure, especially in areas where human cases of LD are known to occur. Lack of experience may have contributed to these results, but even skilled samplers do not detect ticks at each drag. It is also known that sampling the same area twice within hours can give very different results for various reasons (predatorial dynamics, increased tick alertness after first drag, weather variability, etc.) [51]. As null samplings don't necessarily indicate a tick-free environment, and as most sites were sampled only once during the summer, negative sampling results have to be interpreted cautiously, especially near endemic areas. Ideally, samplings could be repeated in the same areas or multiple samplings could be conducted per sites. Perhaps some ambassadors could have benefited from additional practice time during workshops and this could be considered in future ToT initiatives (e.g., classroom role-play, having participants teaching segments to others, etc.). A rating of the ambassador-led outreach and sampling activities could also have provided some insight on ambassadors' integration and mastering of the content, and this should also be integrated in eventual reboots. In spite of the fact that one of the objectives of this project was to collect ticks actively by flannel-dragging, a high number of "passive" ticks–fortuitously encountered on clothes or skin, outside of sampling activities– was also reported during the course of the project (14 encounters, totalling 25 ticks, in three PHUs: Mauricie-Centre-du-Québec, Montérégie, Outaouais). These specimens were not analysed at this stage, but can provide some interesting insight on ambassadors' exposure and behaviour. Indeed, passive ticks represent a good indicator of how much ambassadors and similar workers are exposed to ticks in their daily activities, and how aware they now are of their exposure, having a trained eye for otherwise easily missed arthropods. This points out that, beyond surveillance purposes, ambassador-led tick samplings can be highly valuable in terms of prevention, as witnessing the tick presence first-hand strongly consolidates risk awareness on a local scale. Furthermore, this leads us to believe that passive surveillance could be more effective in collecting ticks in the environment, especially when people are well-informed.

Taking into account that passive tick surveillance is already one of the components of the provincial surveillance program, consideration could be given to exploit this in the future to complement tick datasets, embracing a community or citizen science (CS) approach. CS-based projects, which involve citizens in scientific projects as contributing members at different levels, are praised for their scientific output, data quality and wider societal impact where both professional and citizen scientists benefit from the collaborative effort [52]. Growing in popularity worldwide, the usefulness of CS has already been demonstrated in recent papers, successfully being used in numerous fields of research, from monitoring water quality [53] to biodiversity and invasive alien species [54, 55]. In terms of vector-borne disease, CS has been used to monitor the geographic range expansion of invasive mosquito species (ex. *Aedes albopictus*) in Europe and the US [56–58] or predicting West Nile Virus transmission in North American bird communities [59]. It has also been adopted to look at different aspects of tick-borne diseases in Canada, Finland, the Netherlands and Massachusetts in the recent past [20, 60–66]. On a similar note, and as our work suggests, CS-based tick data could offer significant insights at a scale that is difficult to achieve by a single research group, improving detection resolution. Over time, such data could guide PH interventions and policies at a more local scale.

## Future steps for bigger waves

The ToT-inspired strategy chosen for our project was loosely modeled after the "cascade-training" structure, which has been growing in popularity in a variety of fields in the past as it allows to deploy wide-reaching and long-lasting activities. Cascade-training is typically characterized by a continuous, free flow of knowledge from one level of education to the following, leading to exponential multiplication of learning [67]. Feasibility was demonstrated with this present project in the number of activities that were made possible and the overall number of people reached after a first round of training. However, in order to be properly considered a "cascade", the training developed lacks two main elements: knowledge evaluation (quality control, ideally ongoing, to determine trainer/trainee performance, teaching quality, knowledge retention and changes in behaviour), and network sustainability (ensuring long-lasting results with additional rounds of training) [68]. For instance, ambassadors could have been asked to complete a pre-test and post-test to evaluate their knowledge prior to and after workshops using a knowledge, attitudes, and practices study, and a similar assessment could have been done with their own trainees [69]. Making sure ambassadors and their trainees are able to make waves of their own with the proper information would ensure this program is solid and that it can be effective in the long-term. Another interesting element that should be considered in the future is to compare ambassador tick results with those of "professional flannel draggers", currently in charge of surveillance in the province. Indeed, if PH authorities were to consider implementing a true community-based tick surveillance program, citizen scientists' capabilities to perform samplings according to protocol has to be evaluated in comparison with PH experts. This was not explored in this specific paper and a future project should validate if the two approaches are indeed equivalent quality and quantity-wise.

## Conclusion

Given the sharply increasing risk of LD and other tickborne illnesses in Québec, there is an urgent need to both equip outdoor workers and the general public with tick bite prevention strategies, as well as to continually survey for ticks. By establishing a group of LD education ambassadors that can educate and lead on both fronts, the model presented has the potential to make great strides in protecting people against tick-borne diseases. This project brought to light a first attempt in exploring alternative ways of addressing LD outreach needs in the province of Québec, while concurrently enhancing the scope of the current tick surveillance program. Partnering with citizens has proven a powerful tool in educating communities about the risk of tick-vectored diseases and in encouraging tick bite prevention behaviours. As shown by our work, trained ambassadors have the potential of reaching a large pool of the population while also providing much needed outreach regarding LD risk and prevention in the workplace. We underlined the possibility of combining ToT and CS to potentially build a network of tick collectors and provide a low-cost and efficient methodology for broad tick surveillance with fewer logistical constraints. Data gathered in a similar fashion on a bigger scale could potentially allow for inferences to be made about LD epidemiology. Natural park workers and other targeted ambassadors proved to be trusted sources of information about health risks associated with the recreational use of woodlands, and as such their training has the potential to generate wide-reaching, exponential dissemination of information on LD risk and preventive measures to a relevant target population: workers and visitors of publicly accessible outdoor parks where exposure to LD is increasingly common in Canada. Working against the ticking clock, this exploratory work represents a first step of a possibly bigger process, setting the stage for further initiatives. Additional efforts have to be made to further develop this in

order to create and sustain a potentially ever-growing network of ambassadors in Québec and abroad, generating "more waves downstream of the main waterfall".

## Supporting information

**S1 Table. Workshop evaluation questionnaire and answers.**
(PDF)

**S2 Table. List of ambassador-led activities (sampling and outreach).**
(PDF)

**S3 Table. Pathogen screening in ticks sampled by ambassadors.**
(PDF)

## Acknowledgments

This work would not have been possible without the much-appreciated collaboration of Nicholas Ogden and Sheena Pharand (Public Health Agency of Canada, PHAC). Thank you to François Milord (Direction de santé publique de la Montérégie, Institut national de santé publique du Québec, INSPQ), Geneviève Germain (INSPQ), Catherine Bouchard (PHAC), Tania Abou Chacra (Direction de Santé Publique de l'Estrie) and Patrick Leighton (Université de Montréal) for their expertise and support throughout the project. Thank you to Marie Lemieux (Laboratoire national de santé publique du Québec, LSPQ) for technical assistance with species identification. Thank you to Robbin Lindsay and his team at NML for pathogen screenings. Thank you to Annie Ferland and her colleagues (Nature Conservancy Canada, NCC). Special thanks to Lucas Habib and Sébastien Marcoux (Parks Canada), Marie-Pascale Sassine (INSPQ), Nektaria Nicolakakis (INSPQ), and last but not least, to all ambassadors and other participants that made this project possible.

## Author Contributions

**Conceptualization:** Karl Forest-Bérard, Ariane Adam-Poupart.

**Data curation:** Karl Forest-Bérard.

**Formal analysis:** Karl Forest-Bérard, Karine Thivierge.

**Funding acquisition:** Ariane Adam-Poupart.

**Investigation:** Karl Forest-Bérard, Ariane Adam-Poupart.

**Methodology:** Karl Forest-Bérard, Marion Ripoche, Ariane Adam-Poupart.

**Project administration:** Karl Forest-Bérard, Ariane Adam-Poupart.

**Resources:** Karl Forest-Bérard.

**Supervision:** Karl Forest-Bérard, Ariane Adam-Poupart.

**Validation:** Karl Forest-Bérard, Marion Ripoche, Alejandra Irace-Cima, Karine Thivierge, Ariane Adam-Poupart.

**Visualization:** Karl Forest-Bérard, Ariane Adam-Poupart.

**Writing – original draft:** Karl Forest-Bérard.

**Writing – review & editing:** Karl Forest-Bérard, Marion Ripoche, Alejandra Irace-Cima, Karine Thivierge, Ariane Adam-Poupart.

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
