## [Decision Letter · Decision Letter 0]

9 Feb 2021

PONE-D-20-36491

More than ticking boxes: training Lyme disease education ambassadors to meet outreach and surveillance challenges in Québec, Canada

PLOS ONE

Dear Dr. Forest-Bérard,

Thank you for submitting your manuscript to PLOS ONE. After careful consideration, we feel that it has merit but does not fully meet PLOS ONE’s publication criteria as it currently stands. Therefore, we invite you to submit a revised version of the manuscript that addresses the points raised during the review process.

Both reviewers have experience with education and outreach to reduce risk for Lyme disease.  Both reviewers suggested a substantive revision of the ms.  Reviewer 1's comments in particular should be carefully addressed.

We look forward to receiving your revised manuscript.

Kind regards,

Sam R. Telford III

Academic Editor

PLOS ONE

Journal Requirements:

2. Please include additional information regarding the survey or questionnaire used in the study and ensure that you have provided sufficient details that others could replicate the analyses. For instance, if you developed a questionnaire as part of this study and it is not under a copyright more restrictive than CC-BY, please include a copy, in both the original language and English, as Supporting Information. Moreover, please provide all the training material used, if it is not under a copyright more restrictive than CC-BY.

4. We note that Figures 3 and 4  in your submission contain map images which may be copyrighted. All PLOS content is published under the Creative Commons Attribution License (CC BY 4.0), which means that the manuscript, images, and Supporting Information files will be freely available online, and any third party is permitted to access, download, copy, distribute, and use these materials in any way, even commercially, with proper attribution. For these reasons, we cannot publish previously copyrighted maps or satellite images created using proprietary data, such as Google software (Google Maps, Street View, and Earth). For more information, see our copyright guidelines: http://journals.plos.org/plosone/s/licenses-and-copyright.

(1) You may seek permission from the original copyright holder of Figures 3 and 4 to publish the content specifically under the CC BY 4.0 license. 

5. We note that Figure 1 includes an image of a participant. 

Reviewers' comments:

Reviewer's Responses to Questions

**Comments to the Author**

1. Is the manuscript technically sound, and do the data support the conclusions?

Reviewer #1: Partly

Reviewer #2: Yes

2. Has the statistical analysis been performed appropriately and rigorously? 

Reviewer #1: No

Reviewer #2: N/A

3. Have the authors made all data underlying the findings in their manuscript fully available?

Reviewer #1: No

Reviewer #2: Yes

4. Is the manuscript presented in an intelligible fashion and written in standard English?

Reviewer #1: Yes

Reviewer #2: Yes

5. Review Comments to the Author

Reviewer #1: Thank you for inviting me to review this manuscript. The paper describes an innovative approach to educating those at risk of tick encounters using a “train-the-trainer" or “cascade training” technique. Given the sharply increasing risk of Lyme disease (LD) and other tickborne illnesses in this region of Canada, there is an urgent need to both equip outdoor workers and the general public with tick bite prevention strategies, as well as to continually survey for ticks/diseases. By establishing a group of LD education ambassadors that can educate and lead on both fronts, this program model has the potential to make great strides in protecting people against tickborne disease. I think this manuscript presents an important strategy for community science education/engagement, but I think the scope of the paper needs to be reworked to account for a lack of evaluation and follow-up on the prevention knowledge portion.

Major concerns

While the number and diversity of trainers, and subsequent trainees reached were impressive, I am concerned about the lack of knowledge evaluation at the level of the those trained by the ambassadors, and ultimately whether this program, as described here, can be accurately labels as “cascade training.” A major pitfall of this educational methodology is described using a water metaphor explained by language pedagogy expert Alan Mackenzie (https://alansmackenzie.wordpress.com/2010/02/19/steady-flows-sponges-drips-and-other-wet-patches-a-critical-analysis-of-cascade-training/). This paper does not report whether the correct information and skills were effectively transferred to the participants because they were not surveyed or tested. While feasibility was demonstrated in the number of activities that were made possible, to be deemed successful (as well as properly be considered “cascade”), there must be evaluation (ideally ongoing) of the second stage of implementation to determine trainer/trainee performance and teaching quality and ensure outcome learning/skill goals are being met (Craft, A. (2001). Continuing Professional Development: A Practical Guide for Teachers and Schools. Routledge: London.). Otherwise, this is merely the first step of what could be a complete “cascade training,” but instead describes “sponge” or “trickle down” training where there could be barriers to effective communication that have not yet been identified. What could be changed or improved to better meet the needs of the trainees?

There is no statistical analysis of the survey results from the LD Ambassadors. Please include, at the very least, a frequency distribution/analysis of response data.

I think that the LD ambassador education curriculum is important, scalable, and its description worthy of publication, but the classification of this methodology as “cascade” must be reframed as a program loosely modeled after the “cascade” structure, and clearly define what evaluation still needs to be done in order to be properly considered “cascade.” Including these limitations and next steps are also needed to support the claims that this program can be effective in the long-term.

Given the lack of overall knowledge assessment, I suggest narrowing this paper to focus solely on the effort to equip citizen/community scientists to actively collect ticks on a large scale and not on the dissemination of tick/tick bite prevention information. Collected ticks are objective results that can be directly compared to other published works of citizen/community scientist training efforts.

Specific comments/questions

Line 21-23: Stated goal was “raising awareness about LD risk and prevention among workers and general population,” -- how did you determine changes in attitudes/knowledge? Were pre/post surveys disseminated to the ambassadors and those who they trained?

Line 80-81: Recruiting outdoor workers, due to their increased risk of tick exposure, is a fantastic idea to deeply engage them in the knowledge and practice to protect themselves. Did any of them have prior teaching experience (e.g. park naturalists/interpretive employees)? It might be helpful to have an idea of whether previous educational experience impacted knowledge transmission.

Line 115: What amount of time and proper tools did you anticipate/determine to be necessary to convert trainee to trainer in this scenario?

Line 116: How are you defining "expert?" Expert in teaching, LD prevention/surveillance, or both?

Line 123: What were the evaluation criteria?

Line 202: Was there a minimum number of outreach/sampling activities expected from each?

Line 207: How did the phone call follow-ups ensure long-term retention? Of knowledge? Ambassador participation in the study?

Line 215: Did the structured interviews assess Ambassador knowledge retention or comprehension?

Line 300: Include bar graphs or some other representation (percentages) of survey/interview results from Ambassadors.

Line 320: Without a measure of changes in knowledge/attitudes in the participants reached, it is a reach to claim that knowledge was transferred.

Line 323: What will ensure long-lasting results/prevention of knowledge dilution? How will quality control be performed?

Lines 327-329: What is meant by the "quality of networks" provided by the NCC? And again, how was "success" defined?

Lines 361-362: How does the numbers of ticks collected in these Ambassador-led outings compare to “expert” active tick sampling using the same protocol? It’s challenging to call this a “success” simply because some ticks were found, if in fact the sampling was highly inefficient due to poorly trained community scientists.

Minor comments/typos

Line 57: Change 'has' to 'have': “Changing patterns of emerging or re-emerging vector-borne diseases have been identified...”

Figures/images: I’m sure this is just how the manuscript appears in the review version, but be sure to check the image resolution – the images appear quite blurry on my end.

Reviewer #2: This article entitled “ More thank ticking boxes: training Lyme disease education ambassadors to meet outreach and surveillance challenges in Quebec, Canada” has definitely a merit to be published and written nicely. I do have some minor comments

Major Points:

1. Introduction is too long. Most of the content from line 57-122 should well fit into the discussion and author should give an efforts to do that

2. Cascade Training time seems short to me. Authors should clarify the preventative part more thoroughly. Quebec region has Powassan virus which is fatal. Training component also should provide what other danger could face while they will be working in the field for tick dragging. E.g. how they will tackle if they get attacked by a bear?

3. Since authors found only 11 tick by dragging 28 locations. It seems yield is very low. So more focus should be given on education. Tick collection part should be supplemental

4. This is highly significant that “The CT approach proved to have multiplicative effects on outreach efforts, allowing knowledge to be passed from 321 a single PH professional to nearly 2 000 individuals throughout the south of the province” line 320-321. Authors should add a table if they evaluated their program to highlight their successes.

Minor points:

Abstract

Line 27; used by authorities should be used by Public Health or Professional authorities.

Introduction

Line53: mostly introduced by migratory animals. I think many reports revealed avian species has a significant role as well and need to be mentioned. Ref: 1. Morshed MG, Scott JD, Fernando K, Beati L, Mazerolle DF, Geddes G, Durden LA. Migratory songbirds disperse ticks across Canada, and first isolation of the Lyme disease spirochete, Borrelia burgdorferi, from the avian tick, Ixodes auritulus. J Parasitol. 2005 Aug;91(4):780-90; 2. Morshed MG, Scott JD, Banerjee SN, Banerjee M, Fitgerald T, Fernando K, Mann R, Isaac-Renton J. First isolation of Lyme disease spirochete, Borrelia burgdorferi, from blacklegged tick, Ixodes scapularis, removed from a bird in Nova Scotia, Canada. CCDR 1999; 25(18): 153-155

Materials and Methods

Line 146-148; Training consists of 90 minutes covered project context, pedagogical tools, tick biology, and preventative measures. This seems very short time to me. Trainers definitely should made aware of the rick of carrying out explicitly and need to me mentioned in the articles (mentioned in major points as well).

Conclusion: I didn’t see whether authors have evaluated their program. If yes, should have a table to reflect that and if not author should add a line that project evaluation is underway or will be done

6. PLOS authors have the option to publish the peer review history of their article (what does this mean?). If published, this will include your full peer review and any attached files.

Reviewer #1: No

Reviewer #2: No

---

## [Author Response · Author response to Decision Letter 0]

7 Aug 2021

All comments are addressed in the "Response to Reviewers" file (copied here) : 

Journal Requirements:

Style requirements were revised using PLOS ONE’s templates. 

2. Please include additional information regarding the survey or questionnaire used in the study and ensure that you have provided sufficient details that others could replicate the analyses. For instance, if you developed a questionnaire as part of this study and it is not under a copyright more restrictive than CC-BY, please include a copy, in both the original language and English, as Supporting Information. Moreover, please provide all the training material used, if it is not under a copyright more restrictive than CC-BY.

Copy of the training evaluation questionnaire was included in Supporting Information (in French and English) and additional information was added in corresponding sections of the manuscript (in the “evaluation and analysis” segment), ensuring others could replicate this. The training material itself is under a shared copyright more restrictive than CC-BY as it is projected to be used again in the near future. However, details about the main sources of information are included in the methodology section (in “theoretical component”) so that others could reproduce it in a similar way.

3. We note that you have indicated that data from this study are available upon request. PLOS only allows data to be available upon request if there are legal or ethical restrictions on sharing data publicly. For more information on unacceptable data access restrictions, please see http://journals.plos.org/plosone/s/data-availability#loc-unacceptable-data-access-restrictions. In your revised cover letter, please address the following prompts:

a) If there are ethical or legal restrictions on sharing a de-identified data set, please explain them in detail (e‧g., data contain potentially sensitive information, data are owned by a third-party organization, etc.) and who has imposed them (e‧g., an ethics committee). Please also provide contact information for a data access committee, ethics committee, or other institutional body to which data requests may be sent. 

The minimal anonymized data set necessary to replicate the study findings was uploaded as Supporting Information (ambassador activities data, survey questionnaire and overall answers). Ticks lab analysis (species identification and screening for pathogens) was also added to Supporting Information, although GPS locations were erased for confidentiality reasons. Tick data is a potentially very sensitive information and public health authorities in Quebec do not share it publicly for different reasons, namely because it can negatively affect land owners were the specimens were found (ex. reduce park popularity). To bypass this, INPSQ publishes a LD risk map annually which integrates various surveillance datasets and presents risk at municipality level. All ticks collected by ambassadors were added to the surveillance database used to create the risk map for 2019. This was addressed in our revised cover letter.

4. We note that Figures 3 and 4 in your submission contain map images which may be copyrighted. All PLOS content is published under the Creative Commons Attribution License (CC BY 4.0), which means that the manuscript, images, and Supporting Information files will be freely available online, and any third party is permitted to access, download, copy, distribute, and use these materials in any way, even commercially, with proper attribution. For these reasons, we cannot publish previously copyrighted maps or satellite images created using proprietary data, such as Google software (Google Maps, Street View, and Earth). For more information, see our copyright guidelines: http://journals.plos.org/plosone/s/licenses-and-copyright.We require you to either (1) present written permission from the copyright holder to publish these figures specifically under the CC BY 4.0 license, or (2) remove the figures from your submission:

 (1) You may seek permission from the original copyright holder of Figures 3 and 4 to publish the content specifically under the CC BY 4.0 license. We recommend that you contact the original copyright holder with the Content Permission Form (http://journals.plos.org/plosone/s/file?id=7c09/content-permission-form.pdf) and the following text:

Please upload the completed Content Permission Form or other proof of granted permissions as an "Other" file with your submission. In the figure caption of the copyrighted figure, please include the following text: “Reprinted from [ref] under a CC BY license, with permission from [name of publisher], original copyright [original copyright year].”

USGS National Map Viewer (public domain): http://viewer.nationalmap.gov/viewer/

The Gateway to Astronaut Photography of Earth (public domain): http://eol.jsc.nasa.gov/sseop/clickmap/

Maps at the CIA (public domain): https://www.cia.gov/library/publications/the-world-factbook/index.html and https://www.cia.gov/library/publications/cia-maps-publications/index.html

NASA Earth Observatory (public domain): http://earthobservatory.nasa.gov/

Landsat: http://landsat.visibleearth.nasa.gov/

USGS EROS (Earth Resources Observatory and Science (EROS) Center) (public domain): http://eros.usgs.gov/#

Natural Earth (public domain): http://www.naturalearthdata.com/

After verification, maps shown in Figures 3 and 4 of our manuscript are not copyrighted: they are geographic layers produced by the Government of Québec (open access) and put together by INSPQ’s team at the Bureau d'information et d'études en santé des populations (BIESP). There is no limitation to publish these figures under the CC BY 4.0 license. Additional information was added to the caption of both figures. 

5. We note that Figure 1 includes an image of a participant. 

As per the PLOS ONE policy (http://journals.plos.org/plosone/s/submission-guidelines#loc-human-subjects-research) on papers that include identifying, or potentially identifying, information, the individual(s) or parent(s)/guardian(s) must be informed of the terms of the PLOS open-access (CC-BY) license and provide specific permission for publication of these details under the terms of this license. Please download the Consent Form for Publication in a PLOS Journal (http://journals.plos.org/plosone/s/file?id=8ce6/plos-consent-form-english.pdf). The signed consent form should not be submitted with the manuscript, but should be securely filed in the individual's case notes. Please amend the methods section and ethics statement of the manuscript to explicitly state that the patient/participant has provided consent for publication: “The individual in this manuscript has given written informed consent (as outlined in PLOS consent form) to publish these case details”.If you are unable to obtain consent from the subject of the photograph, you will need to remove the figure and any other textual identifying information or case descriptions for this individual.

Both individuals appearing in Figure 1 have signed the Consent Form for Publication in a PLOS Journal (attached with reviewed manuscript). 

Figure files were uploaded to the Preflight Analysis and Conversion Engine (PACE) digital diagnostic tool. Figures were modified accordingly. 

Reviewers' comments:

Reviewer's Responses to Questions.

1. Is the manuscript technically sound, and do the data support the conclusions?

Reviewer #1: Partly

Reviewer #2: Yes

Response: Having modified the manuscript in light of the reviewers’ comments, we believe the manuscript is now technically sound and stronger than the original version submitted.

2. Has the statistical analysis been performed appropriately and rigorously?

Reviewer #1: No

Reviewer #2: N/A

Response: Given the small size of our study, statistical analysis is indeed limited, but Reviewer #1’s comments and suggestions were taken into account to improve this as much as possible. This exploratory work opens the door to further projects that could collect more data and provide stronger statistical analysis while using a similar methodology. 

3. Have the authors made all data underlying the findings in their manuscript fully available?

The PLOS Data policy requires authors to make all data underlying the findings described in their manuscript fully available without restriction, with rare exception (please refer to the Data Availability Statement in the manuscript PDF file). The data should be provided as part of the manuscript or its supporting information, or deposited to a public repository. For example, in addition to summary statistics, the data points behind means, medians and variance measures should be available. If there are restrictions on publicly sharing data—e‧g. participant privacy or use of data from a third party—those must be specified.

Reviewer #1: No

Reviewer #2: Yes

Response: Taking the journal requirements into consideration, we added information in corresponding sections of the manuscript ensuring others could replicate our work. We also made survey questionnaire and datasets available in Supporting information.________________________________________

4. Is the manuscript presented in an intelligible fashion and written in standard English?

Reviewer #1: Yes

Reviewer #2: Yes

Response: Modifications were made to further improve the quality of the paper. 

5. Review Comments to the Author

N/A

Reviewer #1: 

Thank you for inviting me to review this manuscript. The paper describes an innovative approach to educating those at risk of tick encounters using a “train-the-trainer" or “cascade training” technique. Given the sharply increasing risk of Lyme disease (LD) and other tickborne illnesses in this region of Canada, there is an urgent need to both equip outdoor workers and the general public with tick bite prevention strategies, as well as to continually survey for ticks/diseases. By establishing a group of LD education ambassadors that can educate and lead on both fronts, this program model has the potential to make great strides in protecting people against tickborne disease. I think this manuscript presents an important strategy for community science education/engagement, but I think the scope of the paper needs to be reworked to account for a lack of evaluation and follow-up on the prevention knowledge portion.

We would like to thank Reviewer #1 for their constructive comments and useful insights. This gives us food for thought and further strengthens the manuscript. As illustrated in detail below, we have addressed most of the comments, and justified the few ones we decided not to incorporate in the revised manuscript. 

Major concerns

1.1 While the number and diversity of trainers, and subsequent trainees reached were impressive, I am concerned about the lack of knowledge evaluation at the level of the those trained by the ambassadors, and ultimately whether this program, as described here, can be accurately labels as “cascade training.” A major pitfall of this educational methodology is described using a water metaphor explained by language pedagogy expert Alan Mackenzie (https://alansmackenzie.wordpress.com/2010/02/19/steady-flows-sponges-drips-and-other-wet-patches-a-critical-analysis-of-cascade-training). This paper does not report whether the correct information and skills were effectively transferred to the participants because they were not surveyed or tested. While feasibility was demonstrated in the number of activities that were made possible, to be deemed successful (as well as properly be considered “cascade”), there must be evaluation (ideally ongoing) of the second stage of implementation to determine trainer/trainee performance and teaching quality and ensure outcome learning/skill goals are being met (Craft, A. (2001). Continuing Professional Development: A Practical Guide for Teachers and Schools. Routledge: London.). Otherwise, this is merely the first step of what could be a complete “cascade training,” but instead describes “sponge” or “trickle down” training where there could be barriers to effective communication that have not yet been identified. What could be changed or improved to better meet the needs of the trainees?

We understand and agree with these observations, and we thank Reviewer#1 for pointing this out to us. Our project was exploratory, using a training-of-trainer-inspired methodology to evaluate the feasibility of building the network of ambassadors able to do outreach and sampling activities. This first step sets the stage for a proper cascade training in the future. Our manuscript was rephrased to clarify this and the references mentioned by the reviewer were reviewed and added. 

At the end of the project, we asked ambassadors to provide overall feedback. This final evaluation did highlight that participants “felt better equipped to deal with ticks and the diseases they may carry” after taking part in this project. While this does not count as proper knowledge evaluation, it indirectly shows a certain degree of knowledge retention. As mentioned in the “analysis and evaluation” segment of methodology, we also evaluated indirectly the quality of the implementation of the ToT-inspired approach by assessing different elements (overall number of outreach activities organised per ambassador, number of trainees participating per outreach activity, number of PHUs covered by outreach and tick-sampling activities, number of I. scapularis ticks collected per sampling activity, etc). While this information provides indirect and qualitative insight on the ambassador knowledge and appreciation of the project, it doesn’t allow us to appreciate the actual knowledge retention or potential changes in behaviour. To be considered a proper cascade training and adequately evaluate the results of such approach, we could have evaluated the knowledge and the changes in behavior of the trainers and participants prior to and after the project using a knowledge, attitudes, and practices study (KAPs) (Schotthoefer et al. 2020). For instance, ambassadors could have been asked to complete a pre-test and post-test to evaluate their knowledge before and after workshop, and a similar assessment could have been done with their own trainees. The use of a KAP study was added in the discussion section of the manuscript to address the possible improvements for future projects. 

1.2 There is no statistical analysis of the survey results from the LD Ambassadors. Please include, at the very least, a frequency distribution/analysis of response data.

Given the small size of our study, statistical analyses are indeed limited, but we added the survey questionnaire and descriptive dataset in Supporting information (weighted averages per question on training evaluation). Additionally, note that this exploratory work (proof of concept, validated approach for the province of Québec) is a first step of a possibly bigger process, opening the door to other projects based on a similar approach, gather more information and provide stronger statistical analysis in the future. 

1.3 I think that the LD ambassador education curriculum is important, scalable, and its description worthy of publication, but the classification of this methodology as “cascade” must be reframed as a program loosely modeled after the “cascade” structure, and clearly define what evaluation still needs to be done in order to be properly considered “cascade.” Including these limitations and next steps are also needed to support the claims that this program can be effective in the long-term.

These points are all excellent and relevant. Segments of our discussion were improved to include these elements and clearly indicate what needs to be done along to bring this project further. As further explained in comment 1.1, we transitioned from calling it “cascade training” to “training-of-trainers”, while describing what needs to be done to build a proper CT-training. 

1.4 Given the lack of overall knowledge assessment, I suggest narrowing this paper to focus solely on the effort to equip citizen/community scientists to actively collect ticks on a large scale and not on the dissemination of tick/tick bite prevention information. Collected ticks are objective results that can be directly compared to other published works of citizen/community scientist training efforts. 

We agree that collected ticks are objective results that can be directly compared to other published works of citizen/community scientist efforts, and that the lack of knowledge assessment makes it impossible to accurately evaluate or compare this aspect of our project. However, we believe that the dissemination of tick/tick bite prevention information is non-negligible in terms of public health intervention, since a significant amount of people (nearly 2000 individuals living in LD high-risk areas) were contacted by ambassadors. Among these were several outdoor workers that would not have been otherwise reached by PH authorities based on the current provincial occupational health plan. While we do not have hard data to prove knowledge-retention, we believe these individuals benefitted from some level of added protection from the dissemination of the prevention information. Therefore, we decided to maintain both aspects of the project (outreach and sampling) in this manuscript, with some added nuances such as mentioned in our answer to comment 1.1. 

Specific comments/questions

1.5 Line 21-23: Stated goal was “raising awareness about LD risk and prevention among workers and general population,” -- how did you determine changes in attitudes/knowledge? Were pre/post surveys disseminated to the ambassadors and those who they trained?

This is a valid point. We are aware and we agree that pre/post surveys are essential to quantitively analyse changes in awareness levels, in attitude or knowledge. As explained earlier, in this proof-of-concept project, we only evaluated qualitatively the participants’ perspective at the end of the project only. Most participants stated feeling “better equipped than before” to deal with tick-related issues, which indirectly suggests they are more alert and that they have learned. These elements were added to the discussion of the paper, mentioning that knowledge evaluation should be mandatory for future projects. 

1.6 Line 80-81: Recruiting outdoor workers, due to their increased risk of tick exposure, is a fantastic idea to deeply engage them in the knowledge and practice to protect themselves. Did any of them have prior teaching experience (e‧g. park naturalists/interpretive employees)? It might be helpful to have an idea of whether previous educational experience impacted knowledge transmission.

Some of the recruited workers had prior outreach experience. Since knowledge transmission was not quantitively documented in this project, it is impossible for us to say if previous experience impacted the process. However, since some ambassadors had outreach-related activities as part of their regular work activities throughout summer, we can hypothesize that they were comfortable with the idea of relaying information to others, and that they already had some infrastructure to do so in their organisation. We added information in the manuscript to highlight this in the “ambassador characteristics” section (line 287). 

1.7 Line 115: What amount of time and proper tools did you anticipate/determine to be necessary to convert trainee to trainer in this scenario?

Considering that most workers contacted to be ambassadors were quite busy during tick-activity season and that they were participating outside of their regular work-activities as volunteers, the amount of time available to train new trainers was limited. Determining the duration of each workshop was a question of balance between participants availability and training efficiency. Our selection (180 minutes) was determined to offer a compromise solution with the minimum amount of time necessary for everyone to master the basic aspects of the content presented. The ambassador workbook and its verbatim were made available to everyone to consolidate their learning. In addition to project-specific material, the tools provided were those of the provincial and federal governments. We decided to give in-person workshops (instead of webinars for instance) considering human interaction provides better interaction and a better learning experience. The field segment of each workshop (60-90 minutes spent in wooden paths, explaining and experimenting drag sampling) also highly contributes to learning and raising awareness raising. We added information in the manuscript to clarify this. 

1.8 Line 116: How are you defining "expert?" Expert in teaching, LD prevention/surveillance, or both?

The expert in charge of leading workshops is a professional scientific advisor from the national public health institute of Quebec whose work revolves around LD prevention and/or surveillance, with some level of outreach/pedagogic experience. We clarified this element in the “ambassador training” section of our manuscript. 

1.9 Line 123: What were the evaluation criteria?

This was rephrased as we did not properly evaluate the training. As mentioned in our answer to comment 1.1, our project was exploratory, using a training-of-trainer-inspired methodology to evaluate the feasibility of building the network of ambassadors. 

1.10 Line 202: Was there a minimum number of outreach/sampling activities expected from each?

Ambassadors were invited to lead as many activities as they could during the activity period. No firm minimum was required, but it was suggested that each lead at least 2 sampling activities (2x2000m sampled per ambassadors). We added a segment in the “Ambassador outreach and tick sampling activities” segment of our manuscript to clarify this. 

1.11 Line 207: How did the phone call follow-ups ensure long-term retention? Of knowledge? Ambassador participation in the study?

The follow-up calls aimed to keep everyone mobilized and motivated, ensuring they had everything they needed to move forward with activities. No long-term retention evaluation was performed, and no further tick-related information was given during those calls, but questions were answered and other ambassadors’ stories/experiences were shared. We added a segment in the manuscript to clarify this.

1.12 Line 215: Did the structured interviews assess Ambassador knowledge retention or comprehension?

The semi-structured interviews mainly addressed their overall experience and perception of the project to further improve the approach for future initiatives, but it did not assess knowledge retention. These group interviews were rather informal and provided qualitative information for future improvement of the project as described in our answer to comment 1.1.

1.13 Line 300: Include bar graphs or some other representation (percentages) of survey/interview results from Ambassadors.

The semi-structured group interviews were rather informal and provided qualitative information as described in our answer to comment 1.1. These comments were not compiled and are not presented, but will be useful to guide future projects. 

1.14 Line 320: Without a measure of changes in knowledge/attitudes in the participants reached, it is a reach to claim that knowledge was transferred.

We agree with Reviewer #2 that measure of changes in knowledge/attitudes should have been measured. This comment was addressed in our answer to comment 1.1.

1.15 Line 323: What will ensure long-lasting results/prevention of knowledge dilution? How will quality control be performed?

This is a very good point that we also had in mind. As an exploratory, stand-alone project without perennial funding, long-term mobilisation of trained ambassadors or quality control was unfortunately not possible. However, the tools and documentation provided during the project are still available to ambassadors for future reference, and INSPQ’s experts are also available for remote assistance is needed. Trained ambassadors have shown interest to participate in future activities and could potentially be re-contacted for reboots. 

Similarly: Being able to sample ticks autonomously is useful, but without funding for lab analysis, the amount of information if generates is also limited. To bypass this, we suggested Ambassadors to submit further samples to eTick (www.etick.ca), an online tool developed in collaboration with Bishop University to identify tick species based on pictures only within 24h. The app also provides prevention information (reminders for ambassadors). 

1.16 Lines 327-329: What is meant by the "quality of networks" provided by the NCC? And again, how was "success" defined?

We agree this was ambiguous. The “quality” of NCC’s network is based on its composition: members from conservation-oriented organizations, encompassing several different categories of outdoor workers in risk areas, a public perfect for our project both in terms of outreach and sampling. As explained in the discussion, we consider this partnership to be a success because it did allow us to achieve our objective (to evaluate the feasibility of building the network of ambassadors through a ToT-inspired approach), but most importantly because it gave us access to outdoor park staff, a key interface with the public. Through them, the training has the potential to improve access of the general public to information about LD risks and prevention in a real-world setting (i‧e., visiting a natural park) where this information is highly relevant (see “facilitators and barriers to implementation” segment). Participants’ feedback was also very positive overall. Precisions were added to clarify this in the discussion of our manuscript.

1.17 Lines 361-362: How does the numbers of ticks collected in these Ambassador-led outings compare to “expert” active tick sampling using the same protocol? It’s challenging to call this a “success” simply because some ticks were found, if in fact the sampling was highly inefficient due to poorly trained community scientists.

This is a very solid point, thank you for bringing it to our attention. We agree that, if PH authorities were to consider implementing a community-based surveillance approach in the future, citizen scientists’ capabilities to perform samplings according to protocol has to be evaluated compared to “expert flannel dragger”, usually in charge of surveillance activities in the province. While this was not explored in this specific paper, a future project could explore this and validate if the two approaches are indeed equivalent or not. A segment was added in the last segment of our discussion to clarify this. 

Flannel-dragging is the gold-standard approach specifically validated for collecting Ixodes scapularis ticks in northeastern America, but, like any other fieldwork, its results often vary according the external factors such as weather conditions, period of the day, time of the year, presence of active hosts, etc. This means that a collector (professional or not) might sample the same area twice in a short period of time and find different amount of ticks. It also means that not detecting a tick on any given sampling does not necessarily means the protocol was not properly followed or that the area is completely void of ticks. 

Collected ticks are very important to categorize LD risk areas, which then dictates public health interventions to be taken. Taken in context with LD’s actual epidemiology in Québec, and based on the actual criterions of regional risk assessment in the province, every collected tick by flannel dragging has the potential to change the level of LD risk of the given municipality on the provincial LD risk map of the Institut national de santé publique du Québec. A change in the level of LD risk could translate into different PH interventions and recommendations for this area.

Minor comments/typos

1.18 Line 57: Change 'has' to 'have': “Changing patterns of emerging or re-emerging vector-borne diseases have been identified...”

As suggested by reviewer #2, the introduction segment was shortened and consequently the phrase mentioned here was cropped out.

1.19 Figures/images: I’m sure this is just how the manuscript appears in the review version, but be sure to check the image resolution – the images appear quite blurry on my end.

The compiled submission PDF includes low-resolution preview images of the figures, but special attention was given to figures and images to make sure resolution would not be an issue later on. They were also uploaded to PACE. 

Reviewer #2: 

We would like to thank Reviewer 2 for their positive assessment of the manuscript. As illustrated in detail below, we have addressed most of the comments, and justified the few ones we decided not to incorporate in the revised manuscript.

This article entitled “More thank ticking boxes: training Lyme disease education ambassadors to meet outreach and surveillance challenges in Quebec, Canada” has definitely a merit to be published and written nicely. I do have some minor comments:

Major Points:

2.1 Introduction is too long. Most of the content from line 57-122 should well fit into the discussion and author should give an effort to do that.

We agree with reviewer 2 and efforts were made to shorten the introduction (going from over 120 lines to less than 50). Some of the removed content was shifted in discussion as suggested. 

2.2 Cascade Training time seems short to me. Authors should clarify the preventative part more thoroughly. Quebec region has Powassan virus which is fatal. Training component also should provide what other danger could face while they will be working in the field for tick dragging. E‧g. how they will tackle if they get attacked by a bear?

Workshop duration (90min theory + 90min field exercises) and implementation (one intensive workshop) were determined based on ambassadors’ limited availability. Given the limited time available, some elements had to be prioritized. While focusing on LD, other tick-borne diseases present in the province were briefly mentioned during training (such as Anaplasma phagocytophilum, Babesia microti, Borrelia miyamotoi, etc.). The preventive measures against Ixodes scapularis bites taught to ambassadors also protected them from bites from other tick species such D. variabilis, A. americanum, H. leporispalustris, and I. cookei, which could transmit Powassan virus .

The risk for bear attacks and other natural risks involved when doing fieldwork (e‧g. wild animals, mosquitoes, skin-irritating plants, heat shock, etc.), were not formally addressed during our training, but we did select ambassadors that had a lot of field experience based on their professional backgrounds. Also, sampling protocol required to be executed in pairs or in groups to ensure samplers safety. This information was added to the manuscript in the methodology section as suggested by reviewer#2. 

2.3 Since authors found only 11 tick by dragging 28 locations. It seems yield is very low. So more focus should be given on education. Tick collection part should be supplemental.

We understand that the low amount of ticks collected by ambassadors (n=11) appear low and seems to limit potential interpretations. However, as explained in comment 1.17, collected ticks are very important to categorize LD risk areas, which then dictates public health interventions to be taken. Taken in context with LD’s actual epidemiology in Québec, and based on the actual criterions of regional risk assessment in the province, every collected tick by flannel dragging has the potential to change the level of LD risk of the given municipality on the provincial LD risk map of the Institut national de santé publique du Québec. A change in the level of LD risk could translate into different PH interventions and recommendations for this area. For instance, most ticks collected by ambassadors came from locations were PH authorities already had recorded a presence signal in the past (adding no new information in terms of PH risk). However, one of them came from an area where such signal had not yet been documented, thus changing the risk area for that municipality from 2019 to 2020 (along with other data from the provincial active surveillance program). 

Finally, the fact that ambassadors were indeed able to collect ticks in their environment demonstrates two things related to prevention and education: (1) that the workers recruited for our project are indeed workers at risk of tick bites in their direct environment and that they do benefit from having received educational material of bite prevention; and (2) that the sampling protocol taught is feasible, understandable and effective, but also that they understood when, where and how to find ticks, which speaks volumes in terms of education and awareness.

For these reasons, we decided to maintain the tick collection part in our manuscript, with some added nuances in the discussion. We added information on the limits of flannel-dragging and on the importance of the collected ticks in terms of LD-risk assessment in Québec in our discussion.

2.4. This is highly significant that “The CT approach proved to have multiplicative effects on outreach efforts, allowing knowledge to be passed from a single PH professional to nearly 2 000 individuals throughout the south of the province” line 320-321. Authors should add a table if they evaluated their program to highlight their successes.

The training was not evaluated. Ambassadors were asked complete an evaluation questionnaire about the training after the workshop. The questionnaire used and the overall results have been added to Supplementary information. In addition, at the end of the activity period, ambassadors were asked about their overall perception of the project in a group phone call, qualitatively covering general appreciation, usefulness of the training, perceived autonomy, positive elements of the training, etc. These descriptive comments will be used to guide improvements for future initiatives, but are not presented here. No other evaluation of the project is projected, but a segment on knowledge evaluation and other missing elements has been added in our discussion (as future steps/possible areas of improvements). All these elements were clarified in the corresponding sections of our manuscript. Finally, note that Figure 5 serves the purpose of the table suggested here by reviewer 2 and visually highlight this multiplicative effect. 

Minor points:

Abstract

2.5 Line 27; used by authorities should be used by Public Health or Professional authorities.

Corrected. 

Introduction: 

2.6 Line53: mostly introduced by migratory animals.

I think many reports revealed avian species has a significant role as well and need to be mentioned:

Ref: 1. Morshed MG, Scott JD, Fernando K, Beati L, Mazerolle DF, Geddes G, Durden LA. Migratory songbirds disperse ticks across Canada, and first isolation of the Lyme disease spirochete, Borrelia burgdorferi, from the avian tick, Ixodes auritulus. J Parasitol. 2005 Aug;91(4):780-90; 

Ref. 2: Morshed MG, Scott JD, Banerjee SN, Banerjee M, Fitgerald T, Fernando K, Mann R, Isaac-Renton J. First isolation of Lyme disease spirochete, Borrelia burgdorferi, from blacklegged tick, Ixodes scapularis, removed from a bird in Nova Scotia, Canada. CCDR 1999; 25(18): 153-155

We thank Reviewer#2 for providing additional references. These were reviewed and added to our manuscript. 

Materials and Methods

2.7 Line 146-148; Training consists of 90 minutes covering project context, pedagogical tools, tick biology, and preventative measures. This seems very short time to me. Trainers definitely should be made aware of the risk of carrying out explicitly and need to me mentioned in the articles (mentioned in major points as well).

As explained at comment 2.2, the amount of time available during workshops was limited because of several factors (namely limited availability during summertime and working hours). Although some participants mentioned they would have liked to have a longer workshop and go into more details, most of them could not afford spending more than half a day on this as volunteer work. In a future project, trainings could be designed otherwise to favour quality and long-lasting learning. 

In terms of risk disclaimer, all ambassadors were required to read and sign an informed consent form, which was reviewed for completeness and adequacy by a member of INSPQ’ ethics board. All efforts have been made to protect participants’ privacy, security, and confidentiality (mentioned in the Ethics statement, and in Methodology). Participants recruited for this project were also selected for being experienced outdoor workers, well-equipped to deal with potential risks out in the field. We clarified the segment about this (see Ethics statement). 

Conclusion

2.8 I didn’t see whether authors have evaluated their program. If yes, should have a table to reflect that and if not author should add a line that project evaluation is underway or will be done.

Ambassadors were asked complete an evaluation questionnaire about the training after the workshop. This questionnaire was completed anonymously online (SurveyMonkey). The questionnaire used and the overall results have been added to Supplementary information. After the completion of the project (December 2019), ambassadors could have been contacted for knowledge assessment but we decided not to, given the sanitary context (COVID-19 pandemic). In addition, at the end of the activity period, ambassadors were asked about their overall perception of the project in a group phone call, qualitatively covering general appreciation, usefulness of the training, perceived autonomy, positive elements of the training, etc. These descriptive comments will be used to guide improvements for future initiatives, but are not presented here. No other evaluation of the project is projected, but a segment on knowledge evaluation and other missing elements has been added in our discussion (as future steps/possible areas of improvements). All these elements were clarified in the corresponding sections of our manuscript. 

Thank you to both reviewers for their help.

---

## [Decision Letter · Decision Letter 1]

29 Sep 2021

More than ticking boxes: training Lyme disease education ambassadors to meet outreach and surveillance challenges in Québec, Canada

PONE-D-20-36491R1

Dear Dr. Forest-Bérard,

We’re pleased to inform you that your manuscript has been judged scientifically suitable for publication and will be formally accepted for publication once it meets all outstanding technical requirements.

Kind regards,

Sam R. Telford III

Academic Editor

PLOS ONE

Additional Editor Comments (optional):

Line 342. "Repetitive sampling might mitigate this effect". Not clear why this is a bad thing. If you mechanically remove the ticks and they are no longer there, is that not good? People love to do surveillance but rarely follow up with intervention. You could have ambassadors go back to where they found the ticks and exhaustively resample to remove. After all, if "one tick" could make a difference (Line 410 et seq) the more that are removed in the name of surveillance, the better.

Reviewers' comments:

Reviewer's Responses to Questions

**Comments to the Author**

1. If the authors have adequately addressed your comments raised in a previous round of review and you feel that this manuscript is now acceptable for publication, you may indicate that here to bypass the “Comments to the Author” section, enter your conflict of interest statement in the “Confidential to Editor” section, and submit your "Accept" recommendation.

Reviewer #2: All comments have been addressed

2. Is the manuscript technically sound, and do the data support the conclusions?

Reviewer #2: Yes

3. Has the statistical analysis been performed appropriately and rigorously? 

Reviewer #2: Yes

4. Have the authors made all data underlying the findings in their manuscript fully available?

Reviewer #2: Yes

5. Is the manuscript presented in an intelligible fashion and written in standard English?

Reviewer #2: Yes

6. Review Comments to the Author

Reviewer #2: Authors fulfilled all the questions i raised as well as corrected on all points I mentioned. Under current status on this article is good to go for the print.

7. PLOS authors have the option to publish the peer review history of their article (what does this mean?). If published, this will include your full peer review and any attached files.

Reviewer #2: **Yes: **Dr Muhammad Morshed

---

## [Editor Report · Acceptance letter]

4 Oct 2021

PONE-D-20-36491R1 

More than ticking boxes: training Lyme disease education ambassadors to meet outreach and surveillance challenges in Québec, Canada 

Dear Dr. Forest-Bérard:

I'm pleased to inform you that your manuscript has been deemed suitable for publication in PLOS ONE. Congratulations! Your manuscript is now with our production department. 

Kind regards, 

on behalf of

Dr. Sam R. Telford III 

Academic Editor

PLOS ONE